# Effect of Grass Carp Scale Collagen Peptide FTGML on cAMP-PI3K/Akt and MAPK Signaling Pathways in B16F10 Melanoma Cells and Correlation between Anti-Melanin and Antioxidant Properties

**DOI:** 10.3390/foods11030391

**Published:** 2022-01-29

**Authors:** Zizi Hu, Xiaomei Sha, Lu Zhang, Sheng Huang, Zongcai Tu

**Affiliations:** 1National R&D Center for Freshwater Fish Processing, College of Chemistry and Chemical Engineering, Jiangxi Normal University, Nanchang 330022, China; Hzz13667949044@163.com (Z.H.); zhanglu00104@163.com (L.Z.); 2College of Life Science, Jiangxi Normal University, Nanchang 330022, China; shaxiaomei1987@sina.com; 3State Key Laboratory of Food Science and Technology, Nanchang University, Nanchang 330047, China; hsheng21@126.com

**Keywords:** collagen peptide, B16F10 cells, melanogenesis, tyrosinase, antioxidant

## Abstract

Peptide Phe-Thr-Gly-Met-Leu (FTGML) is a bioactive oligopeptide with tyrosinase inhibitory activity derived from gelatin hydrolysate of grass carp scales. Previous studies have shown that FTGML addition can effectively inhibit mushroom tyrosinase activity in vitro, and also has some effect on the inhibition of melanogenesis in zebrafish in vivo, but the underlying mechanism is not fully understood. In this study, we used FTGML to treat B16F10 melanoma cells, and found a significant inhibition of tyrosinase activity and melanin synthesis. Interestingly, the treatment showed a strong correlation between antioxidant activity and anti-melanin, which was associated with FTGML reducing the involvement of reactive oxygen species in melanin synthesis. Furthermore, FTGML reduced melanogenesis in B16F10 cells by downregulating the cAMP-PI3K/Akt and MAPK pathways (p38 and JNK). These results suggested that FTGML can reduce melanin production in mouse B16F10 melanoma cells through multiple pathways.

## 1. Introduction

Melanin is a macromolecule biological pigment, mainly composed of eumelanin and pheomelanin. Eumelanin is brown-black, and pheomelanin is red-brown. The different proportions of the two pigments cause the different colors of hair and skin [1]. Melanin can protect the skin from the harmful effects of environmental pollution factors, such as ultraviolet radiation and oxidative stress. However, excessive accumulation of melanin can lead to the appearance of many skin diseases, including melasma, freckles, age spots, and other pigmentation syndromes. The process of melanin synthesis is extremely complicated, mainly involving three enzymes in the tyrosinase gene family, namely tyrosinase (Tyrosinase, TYR), tyrosinase-related protein-1 (Tyrosinase-related protein-1, TRP-1), and tyrosinase-related protein-2 (Tyrosinase-related protein-2, TRP-2), in which tyrosinase is the key enzyme in this reaction [2].

It Is well known that melanocortin 1 receptor (MC1R)/α-melanocyte, stimulating hormone (α-MSH), phosphatidylinositol 3-kinase (PI3K)/activation of phosphorylation of protein kinase B (Akt), mitogen-activated protein kinase (MAPK), Wingless/Integrated (Wnt)/β-catenin, nitric oxide (NO), and other signaling pathways are involved in melanin production of melanocytes [3,4]. Microphthalmia-associated transcription factor (MITF) are the most important molecular targets in these pathways, and the change of MITF expression is directly related to abnormal skin and hair pigments [5]. MITF is a basic helix-loop-helix-leucine zipper transcription factor, which can share a highly conserved sequence with the m-box motif of the promoter region (TYR, TRP-1, TRP-2 in the promoter region; that is, 5′-AGTCATGTGCT-3′), and regulates the expression of TYR, TRP-1, and TRP-2; thus, regulating the production of melanin [6]. Previous studies have found that some peptides inhibit melanin synthesis mainly through ERK in the mitogen-activated protein kinase (MAKP) signaling pathway [7,8]. In the previous experiments of this study, we found a peptide Phe-Thr-Gly-Met-Leu (FTGML) could promote cell apoptosis, which was related to the other two pathways of MAKP (i.e., p38 and JNK) [9], but there was no relevant report that the peptide could inhibit melanin synthesis through these two pathways.

In recent years, people have tried to develop new natural-derived whitening cosmetics or skin bleaching agents, as natural products have the advantages of being less toxic and having fewer side effects compared to chemicals. Examples include tea [10], mung bean seed [2], etc. Fucoidan derived from hizikia fusiforme can significantly inhibit the expression pathway of tyrosinase and tyrosinase-related proteins in B16F10 cells by regulating the extracellular signal-regulated kinase mitogen-activated protein kinase (ERK-MAPK), and downregulating the MITF [11]. Tea catechins downregulate MITF expression by inhibiting cyclic adenosine monophosphate (cAMP), leading to subsequent phosphorylation of CREB, and a decrease in the levels of tyrosinase, TRP-1, and TRP-2, thereby reducing melanin synthesis [12]. However, at present, most research on natural products that inhibit melanin mostly focus on plant sources. Natural products from animal sources, especially fish, have less research in this area, and most of them focus on basic index evaluation. The previous research of our group found that grass carp fish scale gelatin hydrolysate has a good whitening effect, and a new peptide consisting of five amino acids (phenylalanine-threonine-glycine-methionine-leucine, FTGML) was obtained, with good in vitro tyrosinase inhibitory activity (IC_50_ value was 1.89 mM). However, the melanin inhibitory mechanism is still unclear.

In addition, there are reports that hydrogen peroxide (H_2_O_2_) and other ROS and RS are produced during melanogenesis, leading to advanced oxidative stress in melanocytes. Therefore, the use of free radical scavengers in this process can play a role in feedback regulation. Studies have shown that ROS scavengers and ROS generation inhibitors may inhibit ultraviolet-induced melanogenesis [13]. Antioxidants, such as ascorbic acid derivatives and reduced glutathione (GSH), have also been used as inhibitors of melanin production [14,15]. Therefore, it is necessary to pay attention to the antioxidant effect and melanin inhibitory effect of compounds or natural products at the same time, and explore the relationship between the two effects, but there are few studies on the relevant mechanism.

In terms of cell structure and melanin synthesis, the mouse melanoma cell line B16F10 is highly consistent with human melanoma cell. Furthermore, it is difficult to culture human melanoma cells when evaluating the efficacy of active whitening, due to its stricter requirements for cell culture and handling condition. The mouse melanoma cell line B16F10 has the advantages of multiple passages, rapid development, relatively simple culture conditions, high malignancy, and good tumorigenicity. As such, B16F10 cells are widely used as the effective cells for cell evaluation of whitening activity substances [16,17,18]. In this study, we aim to further analyze the anti-melanin effect and potential mechanism of FTGML based on the previous stage in the mouse melanoma cell line B16F10. It is also assumed that FTGML functionally causes cAMP-PI3K/Akt- and MAPK-mediated downregulation of MITF, which is the main cascade of melanin production, leading to the decrease of tyrosinase in melanocytes. In addition, we also studied the correlation between anti-melanin and anti-oxidation in melanocytes, and provided the possibility of predicting its anti-melanin potential through antioxidant activity in the future. Our research contributes to the development of new cosmetic ingredients, food supplements, and functional foods containing collagen peptides.

## 2. Materials and Methods

### 2.1. Experimental Materials

FTGML was obtained from the grass carp fish scale gelatin hydrolysate. In brief, grass carp fish scale gelatin was hydrolyzed by alcalase and gastrointestinal simulated digestion to obtain a mixture of peptides. The peptides with tyrosinase inhibitory activity were screened and identified by bioaffinity ultrafiltration-mass spectrometry, and peptide FTGML showed the best tyrosinase inhibitory activity. FTGML was synthesized in Shanghai Sangon Biotechnology Co., Ltd. (Shanghai, China). The purity of all synthetic peptides was above 95%. Mushroom tyrosinase (EC1.14.18.1, with specific activity 6680 U/mg) and L-DOPA were produced by Sigma-Aldrich (St. Louis, MO, USA). The murine melanoma B16F10 cells (CL0039) were acquired from Fenghui Biological Technology Co., Ltd. (Changsha, China). All other reagents were analytical grade.

### 2.2. Cell Culture

B16F10 cells was cultured by the medium of RPMI-1640, with 10% fetal bovine serum (FBS), and 1% penicillin/streptomycin at 37 °C in 5% CO_2_.

### 2.3. Cell Viability Assay

Cell viability was assessed by Cell Counting Kit-8 (CCK-8) assay. The cells in the logarithmic growth phase were digested by trypsin, and counted under the microscope to make a cell suspension of 1~5 × 10^4^ cells/mL. A 100 µL cell suspension was taken to a 96-well culture plate with 1~5 × 10^3^ cells/well. Wells added with 100 µL medium were used as blank control, and wells added with different concentrations (0, 0.1, 0.2, 0.4, 0.8, 1.6 mg/mL) of FTGML solution and 0.75 mg/mL of kojic acid were used as the experimental groups and positive control groups, respectively. After 0 h, 24 h, and 48 h of incubation at 37 °C, the 1:10 volume ratio mixed CCK-8 and serum-free essential basic medium was added to the test well at 100 µL per well, and incubated for 1 h at 37 °C in a 5% CO_2_ incubator. The absorbance at 450 nm was measured with a microplate reader.

### 2.4. Determination of Apoptosis Rate of B16F10 Cells

Cells in 2.3 were collected and performed according to the Annexin V-FITC kit (Thermo Fisher Scientific, Carlsbad, CA, USA). The apoptosis of B16F10 cells was analyzed by flow cytometry (Ex = 488 nm, FL1 Em = 525 ± 20 nm, FL2 Em = 585 ± 21 nm) and FlowJo software.

### 2.5. Assay of Tyrosinase Activity

Tyrosinase activity was measured using a Tyrosinase activity assay kit (BC4055, Beijing Solarbio Science & Technology Co., Ltd., Beijing, China). OD was measured at 475 nm.

### 2.6. Measurement of Melanin Synthesis

B16F10 cells were seeded in 6-well plates at a density of 2 × 10^6^ cells/well. After incubation for 24 h, the cells were treated with FTGML (0–1.6 mg/mL) or kojic acid (0.75 mg/mL), and incubated at 37 °C in 5% CO_2_ for 48 h. The cells were washed with PBS, digested with trypsin, and centrifuged at 1500 rpm for 10 min. To dissolve the cell precipitates, 1 mL 1 M NaOH containing 10% DMSO was used at 80 °C for 1 h. The optical density of each well at 405 nm was measured with a microplate reader, and the content of melanin in the cell was calculated [12]:(1)$$\text{Melanin}\;\text{content}/\%=\frac{(A_{s}-A_{b})}{(A_{c}-A_{b})}\;\times \;100\%$$
where, A_s_ is the absorbance of experimental wells; A_c_ is the absorbance of control well; A_b_ is the absorbance of blank wells.

### 2.7. Assay of Antioxidant Activity

The centrifuge precipitate in Section 2.6 was dissolved with 1 mL 1 M NaOH containing 10% DMSO at 80 °C for 1 h were collected separately for antioxidant activity determination. The content of reduced glutathione (GSH), oxidized glutathione (GSSG), reactive oxygen species (ROS), and malondialdehyde (MDA), and the activity of superoxide dismutase (SOD), catalase (CAT), and glutathione peroxidase (GPX) were measured using different assay kits that were all from Beijing Solarbio Science & Technology Co., Ltd., Beijing, China, and the item numbers were as follows: GSH by BC1175; GSSG by BC1185; ROS by CA1410; MDA by BC0025; SOD by BC0175; CAT by BC0205; GPX by BC1195.

### 2.8. Measurement of Intracellular cAMP Concentration

B16F10 cells were inoculated in 6-well plates at a density of 2 × 10^5^ cells/well. After 24 h incubation, the cells were treated with FTGML (0–1.6 mg/mL) or kojic acid (0.75 mg/mL), and incubated at 37 °C in 5% CO_2_ for 48 h. Cells were washed with cold PBS, and dissolved on ice in a RIPA buffer containing protease and phosphatase inhibitors. The samples were centrifuged at 1000× *g* for 20 min to remove impurities and cell debris, and the supernatant was used for cAMP detection. A cAMP immunoassay kit (Jiancheng Bioengineering Institute, Nanjing, China) was used to measure intracellular cAMP concentration [12].

### 2.9. Western Blot Analysis

B16F10 cells were treated with FTGML (0–1.6 mg/mL) or kojic acid (0.75 mg/mL) for 48 h. Lysis buffer (150–250 μL) was added to each well to complete the lysis. Then, the lysed samples were centrifuged at 12,000× *g* for 15 minutes at 4 °C, and the supernatant was obtained for protein quantification (BCA assay kit, PICPI23223, Thermo Fisher Scientific, Carlsbad, CA, USA), and stored in a refrigerator at −80 °C for the next analysis.

An equal amount of protein (20 μg/sample) was bathed in boiling water for 10 min after being mixed with 5 × SDS loading buffer, separated by 12% SDS polyacrylamide gel electrophoresis, and transferred to a polyvinylidene fluoride (PVDF) membrane (Thermo Fisher Scientific, Carlsbad, CA, USA). The PVDF membrane was blocked in 5% skim milk in TBST buffer (PBS containing 0.05% Tween-20) for 1 h at room temperature. The membranes were incubated with several antibodies overnight at 4 °C after a short wash. These antibodies include anti-MITF (1:500), anti-GAPDH (1:2500), anti-p38 (1:1000), anti-p -p38 (1:1000), anti-p-JNK (1:200), anti-JNK (1:500), anti-p-PI3K (1:500), anti-PI3K (1:500), anti-p-Akt (1:2000), and anti-Akt (1:1000). After incubation, the membrane was washed thoroughly with TBST buffer, and further incubated with goat anti-rabbit HRP-labeled secondary antibody (1:1000, Beyotime Biotechnology Co., Ltd., Shanghai, China) at 37 °C for 1 h. The blots were visualized using enhanced chemiluminescence (ECL), and quantified by Image-pro plus software (Media Cybernetics, Rockville, MD, USA).

### 2.10. Statistical Analysis

All results were expressed as mean ± standard deviation. SPSS version 20 (SPSS Inc., Chicago, IL, USA) was used for analysis of variance (*p* < 0.05). The Duncan multi-range test was used for the comparison of means. Three replicates were used for each analysis. Principal component analysis (PCA) was used to analyze the correlation between the different properties of anti-melanin and anti-oxidation. The above indicators were used as active variables, and the concentration of FTGML was used as the observed value. Two-dimensional or three-dimensional graphs were drawn using Origin Pro 2018 software. Hierarchical cluster analysis (HCA) was used to visualize and emphasize the similarities between individuals. The difference is usually expressed by the distance between individuals [19]. Origin Pro 2018 software was used to determine the correlation between indicators by the Pearson correlation coefficient in binary linear correlation.

## 3. Results and Discussion

### 3.1. Effects of FTGML on B16F10 Cells Viability

B16F10 cells were treated with 0.1–1.6 mg/mL FTGML and 0.75 mg/mL kojic acid for 0 h, 24 h, and 48 h, respectively. The cell viability is expressed as a percentage relative to the cells in the blank control group (cells without any drugs), and the results are shown in Figure 1. FTGML and kojic acid showed low toxicity to B16F10 cells at all treatment concentrations (cell viability > 80%, [20]). Based on these results, the concentration of FTGML in the range of 0.1 mg/mL to 1.6 mg/mL can be used for subsequent experiments.

### 3.2. Effects of FTGML on Apoptosis Rate of B16F10 Cells

Apoptosis is a basic biological phenomenon of cells, and plays an essential role in the removal of unwanted or abnormal cells by multicellular organisms. The disorder of apoptosis may be directly or indirectly related to the occurrence of many diseases, such as tumors and autoimmune diseases. Therefore, an apoptosis assay is often used to evaluate the development and application potential of active ingredients in food in the field of functional food [21].

The effect of 0.1~1.6 mg/mL FTGML treatment for 48 h on the apoptosis of B16F10 cells is shown in Figure 2. It can be seen from Figure 2A that as the concentration of FTGML increased, the proportion of normal living cells gradually decreased, and the proportion of early apoptotic cells, late apoptotic cells, and necrotic cells increased. Figure 2B shows the apoptosis rate of B16F10 cells (i.e., the total proportion of early apoptotic cells to late apoptotic cells). It can be seen from the figure that FTGML treatment promoted the apoptosis of cells, and the apoptosis rate was positively correlated with the concentration of FTGML. There was no significant difference in apoptosis rate between B16F10 cells treated with kojic acid (0.75 mg/mL) and FTGML at high concentration (≥0.8 mg/mL) (*p* > 0.05).

STAT3 (signal transducer and activator of transcription 3) is both a cytoplasmic signal molecule and a nuclear transcription factor, which is involved in cell proliferation, transformation, and migration. At present, STAT3 has been identified as a major oncogene in the development of melanoma [22]. There is also genetic evidence for a direct role of STAT3 in melanoma cell transformation [23]. As can be seen from Figure 2C, when the concentration of FTGML was greater than 0.4 mg/mL, the ratio of phosphorylated STAT3 (p-STAT3) to STAT3 was lower than 1, and significantly lower than that of the blank group (*p* < 0.05). This suggests that medium-high concentrations (≥0.4 mg/mL) of FTGML can reduce the activation/phosphorylation of STAT3. Some researchers have found that increasing the activation/phosphorylation level of STAT3 can promote the growth of melanoma, whereas silencing STAT3 can significantly inhibit the proliferation of melanoma cells, and promote cell apoptosis [22]. This is consistent with the results shown in Figure 2B.

### 3.3. Effects of FTGML on Intracellular Melanogenesis

The first step in studies of melanin production is usually to measure intracellular tyrosinase activity, since tyrosinase is the rate-limiting enzyme involved in melanin synthesis [12]. To test the inhibitory effect of FTGML on intracellular tyrosinase activity, B16F10 cells were exposed to FTGML solution in the concentration range of 0~1.6 mg/mL for 48 h. As shown in Figure 3A, FTGML significantly inhibited tyrosinase activity compared with untreated cells (*p* < 0.05). With the increase of FTGML concentration, intracellular tyrosinase activity decreased gradually. When the FTGML concentration was 1.6 mg/mL, the residual tyrosinase activity was 43.87 ± 5.89% of that of the control. This was not a statistically significant difference (*p* > 0.05) from kojic acid (positive control substance) at 0.75 mg/mL.

To further explore the effect of FTGML on melanin production, we detected the residual melanin content in B16F10 cells after FTGML treatment. The results in Figure 3B show that FTGML significantly reduced melanin in B16F10 cells, especially when the FTGML concentration was higher than 0.8 mg/mL. When the concentration of FTGML was 1.6 mg/mL, the intracellular melanin content was 56.44 ± 15.05% of the control group. The content of residual melanin in cells after kojic acid (0.75 mg/mL) treatment was 46.67 ± 6.31% of that of the control group, which was not significantly different from that after FTGML (1.6 mg/mL) treatment (*p* > 0.05). These results suggest that FTGML can achieve whitening by inhibiting the synthesis of melanin in cells, and can achieve a similar effect to kojic acid to a certain extent. This is similar to the results of Han et al., who found that oyster hydrolysate significantly reduced intracellular melanin content [24].

### 3.4. Effects of FTGML on Intracellular Antioxidant Activity

In the production of melanin, tyrosine is required to produce dopamine in an oxidized environment and subsequently, to produce dopamine. Therefore, the existence of reactive oxygen species, such as superoxide anion and hydroxyl radical, is conducive to the synthesis of melanin [25]. Studies have shown that peptides with oxygen free radical scavenging ability can inhibit the biosynthesis of melanin in cells [8,26].

Glutathione (GSH) is an important intracellular regulatory metabolite, and its redox states (i.e., reduced glutathione (GSH) and oxidized glutathione (GSSG)) are important for many physiological processes. The effect of FTGML on GSH content in B16F10 cells is shown in Figure 4A. The content of GSH in cells is positively correlated with the concentration of added FTGML. When the concentration of FTGML is 1.6 mg/mL, the content of GSH in cells is 1.78 times higher than that in the blank group. GSH, as a small peptide substance with strong reducibility in cells, can scavenge intracellular free radicals, leading to its critical role in the production of melanin [27]. The effect of FTGML on GSSG content in B16F10 cells is shown in Figure 4B. The addition of FTGML reduced the content of GSSG in B16F10 cells. The GSG:GSSG ratios of B16F10 cells treated with different concentrations of FTGML (0, 0.1, 0.2, 0.4, 0.8, and 1.6 mg/mL) and 0.75 mg/mL kojic acid were 2.10 ± 0.23, 2.57 ± 0.09, 3.61 ± 0.10, 4.79 ± 0.25, 5.00 ± 0.45, 5.22 ± 0.29, 5.68 ± 0.22, respectively. These results indicate that FTGML can destroy the oxidative environment in cells. It can maintain the reductive power in cells, and regulate the level of GSH in cells, thus inhibiting the production of melanin.

It has been shown that ROS play an important role in the regulation of melanogenesis and melanocyte proliferation [28]. ROS scavengers or formation inhibitors can reduce melanogenesis in melanocytes. Malondialdehyde (MDA) is one of the products of cellular membrane lipid peroxidation, and can be used to reflect the extent of oxidative stress damage to cells [29]. To elucidate the protective mechanism of FTGML against oxidative stress in B16F10 cells, the intracellular ROS level and MDA content of B16F10 cells were measured, as shown in Figure 4C,D, respectively. Intracellular ROS and MDA dropped to 62.81% and 50.46% of the untreated group, respectively, as the amount of FTGML was increased. After kojic acid treatment, intracellular ROS and MDA dropped to 58.94% and 51.49% of the untreated group, respectively. This means that after treatment with FTGML (1.6 mg/mL) and kojic acid (0.75 mg/mL), there was no significant difference (*p* > 0.05). This demonstrated that FTGML significantly inhibited intracellular ROS and MDA production, and protected B16F10 cells from oxidative damage. This is similar to the study by Huang et al., who found that [8]-gingerol inhibited melanogenesis in melanoma cells, and that the addition of 100 μM reduced intracellular ROS levels to 71.01 ± 1.45% of the blank group, indicating that it significantly depleted ROS levels in B16F10 cells [30].

SOD, CAT, and GPx are antioxidant enzymes that work together to decrease ROS, and protect cells from oxidative stress damage [31]. It can be seen from Figure 4E,G that the activities of SOD, CAT, and GPx increased by 63.10%, 64.53%, and 69.29%, respectively, after FTGML treatment of B16F10 cells, and increased by 68.51%, 56.00%, and 84.33%, respectively, after treatment with kojic acid. Compared with kojic acid, similar effects to FTGML were observed in SOD and CAT activities (*p* > 0.05). However, there is still a certain gap between the two in GPx activity (*p* < 0.05), and with the increase of FTGML concentration, there is no significant change in GPx activity (*p* > 0.05). The activity of antioxidant enzymes (SOD, CAT, and GPx) significantly affects the sensitivity of the skin to oxidative damage, including skin pigmentation problems [32]. Therefore, FTGML shows a protective effect on oxidation by increasing the activity of antioxidant enzymes in B16F10 cells, thereby avoiding melanin deposition. Some researchers’ reports also indicate that peptides interfere with skin biochemical reactions by protecting cells against oxidative damage, for example, sorghum kafrins-derived peptide fraction [26], and this is mainly due to their ability to enhance the activity of antioxidant enzymes in the cell.

### 3.5. Effects of FTGML on the Melanogenesis-Related Signaling Pathway in B16F10 Melanoma Cells

In this study, we examined the expression of melanogenesis-related proteins, including MITF and cAMP, and melanogenesis-regulating molecules, including PI3K/AKT, p38, and JNK, to elucidate the potential mechanisms by which FTGML inhibits melanogenesis in B16F10 melanoma cells. The results are shown in Figure 5.

cAMP is a well-known intracellular second messenger, and its mediated signaling is the main cascade of melanin production, which is largely influenced by changes in intracellular cAMP levels [12]. As shown in Figure 5A, FTGML (1.6 mg/mL) and kojic acid (0.75 mg/mL) reduced intracellular cAMP levels in B16F10 cells by 48.77% and 45.38%, respectively, compared to untreated controls (*p* < 0.05). cAMP is an important mediator of intracellular signal-activated protein kinase A (PKA). In the absence of cAMP, PKA is inactive, and its two PKA catalytic subunits bind to the two PKA regulatory subunits. When cAMP is present, cAMP binds to the regulatory subunits of PKA, and induces dissociation of the PKA catalytic subunits from the PKA holoenzyme complex, and the released PKA catalytic subunits are activated, ultimately activating the cAMP response element binding protein (CREB), and thus promoting melanin synthesis [33]. Therefore, it can be concluded that FTGML treatment reduces the intracellular cAMP level, which is beneficial to the regulation of subsequent signaling pathways. Similar results were found by Han et al. Oyster hydrolysate exhibited anti-melanogenic activity by downregulating the cAMP signaling pathway, thereby reducing melanin synthesis [24]. Phosvitin-derived peptide Pt5 was shown to be involved in the cAMP pathway to inhibit melanogenesis, with no significant effect on the Wnt and MAPK signaling pathways [34].

cAMP can also regulate melanogenesis through PKA non-dependent mechanisms, such as phosphatidylinositol-3-kinase (PI3K). One of the key effectors of PI3K is protein kinase B (Akt). The results shown in Figure 5C were obtained by analysis of Figure 5B. Both FTGML and kojic acid treatment decreased the expression of p-PI3K and p-Akt compared to untreated controls. p-PI3K/PI3K and p-Akt/Akt ratios of B16F10 cells were 0.67 and 0.89, respectively, after 1.6 mg/mL FTGML treatment. In addition, 0.75 mg/mL kojic acid treatment decreased the p-PI3K/PI3K and p-Akt/Akt ratios to 0.59 and 0.77, respectively. Under the stimulation of external signals and intracellular cAMP, activated Akt enhances the binding of MITF to the M-box by phosphorylating glycogen synthase kinase 3β (GSK3β), and promotes its loss of activity. The reduction in GSK3β activity enhances the binding of MITF to the M-box, and synergistically stimulates the tyrosinase promoter with MITF, enhancing its binding to the tyrosinase promoter, and thus, promoting melanogenesis [35]. The results showed that FTGML reduced the phosphorylation level of PI3K/Akt by downregulating cAMP. It has been shown that diosgenin has the effect of regulating the reduction of PI3K expression and phosphorylation of AKt, which affects the phosphorylation of downstream GSK-β, which, in turn, allows the downregulation of MITF expression, leading to a reduction in melanin synthesis [36].

In addition to the cAMP-mediated signaling pathway described above, there is also the mitogen-activated protein kinase (MAPK) signaling pathway, which is involved in the expression and activation of MITF [37]. Based on the apoptotic results in Section 3.2, we speculate that it may be FTGML that promotes apoptosis in B16F10 cells, resulting in reduced melanin. Several past studies have shown that JNK and p38 are the two most important factors in MAPK that affect apoptosis [9,38,39]. As shown in Figure 5C, both FTGML and kojic acid treatment reduced the expression of p-JNK and p-p38 compared to untreated controls. p-JNK/JNK and p-p38/p38 ratios of B16F10 cells after 1.6 mg/mL FTGML treatment were 0.65 and 0.70, respectively. In addition, 0.75 mg/mL kojic acid treatment reduced the p-JNK/JNK and p-p38/p38 ratios of B16F10 cells, which had p-JNK/JNK and p-p38/p38 ratios of 0.68 and 0.63, respectively. It has been reported that the inhibition of the JNK pathway reduces melanin production [40], and that phosphorylation of p38 activates MITF expression and upregulates melanogenesis-related proteins, which, in turn, affects melanin synthesis [41]. These results suggest that FTGML inhibits melanin synthesis through downregulation of the JNK and p38 signaling pathways.

MITF is located downstream of these signaling pathways and upstream of tyrosinase, and is involved in the activation of several melanocyte-related genes. It regulates both tyrosinase and its associated proteins on melanin synthesis, as well as the growth, development, and differentiation of melanocytes [42]. As shown in Figure 5D, the expression of MITF was significantly reduced after FTGML acted on melanocytes, and the inhibition was most evident at the highest concentration of 1.6 mg/mL of FTGML, which was consistent with our prediction, indicating that FTGML could reduce the expression of MITF. It was also verified that inhibition of the above signaling pathways (cAMP-PI3K/Akt and MAPK pathways) also significantly affected the expression of MITF. Based on these results, we hypothesize that FTGML downregulates the phosphorylation of PI3K/Akt in this signaling pathway by decreasing cAMP levels. It also downregulates the phosphorylation of JNK and p38 in the MAPK signaling pathway. Together, these two pathways are involved in FTGML’s inhibition of MITF expression-mediated melanogenesis, with the PI3K signaling pathway being relatively more important. Of the available studies on tyrosinase inhibitory peptides, very little has been done on their signaling pathways. Peptides derived from the fermented microalga (*Pavlova lutheri*) [8] and oyster hydrolysate [24] have been demonstrated, but they have all been studied by selecting one of the signaling pathways. The findings of the present study suggest the possibility that multiple signaling pathways may still act together in melanin inhibition.

### 3.6. Principal Component (PCA), Cluster Analyses (HCA), and Correlation Analysis

Principal component analysis was performed on 10 indicators from 6 samples with different FTGML additions using SPSS 20.0 software. As shown in Table 1, a total of two principal components were extracted, and the cumulative value of the contribution of the principal components reached 94.836%, which can explain the vast majority of the original information. Principal component 1 is the most important, with a contribution of 87.144% of the variance, which can represent 87.144% of the total information. Principal component 2 has a variance contribution of 7.692%.

The corresponding eigenvectors and load matrices of the principal components are shown in Table 2. It can be concluded that PC1 is associated with 15% melanin, 17% cAMP, 15% GSH, 26% CAT, 23% MDA, 15% ROS, and 44% SOD, whereas PC2 is mainly associated with 34% tyrosinase, 44% GPx, and 57% GSSG. Using PC1 as the *x*-axis, and PC2 as the *y*-axis, a plot was drawn based on the corresponding load values, as shown in Figure 6A. Ten indicators were scattered in the first and third quadrants of the axes. The linear equation of the combined score of each principal component was derived from the eigenvector matrix of the two principal components, and the relative contribution of the variance corresponding to each principal component was used as the weight to build the comprehensive evaluation model: F1 = 0.154 × 1 + 0.170 × 2 − 0.154 × 3 − 0.146 × 4 − 0.366 × 5 − 0.259 × 6 + 0.238 × 7 + 0.225 × 8 + 0.147 × 9 − 0.442 × 10; F2 = −0.011 × 1 − 0.024 × 2 + 0.342 × 3 − 0.001 × 4 + 0.567 × 5 + 0.130 × 6 − 0.436 × 7 − 0.089 × 8 + 0.005 × 9 + 0.372 × 10. Based on the 53.110% of principal component 1, and 41.727% of principal component 2, the composite score function can be derived as F = 0.53110 × F1 + 0.41727 × F2. The composite score of each sample was calculated and ranked by the above model (Table 3). The F1 and F2 scores for each sample are represented as coordinates in Figure 6A. Combined with the PCA composite evaluation model scores, the samples were further classified into four groups by hierarchical cluster analysis (HCA) (Figure 6B-1): control (0), low performing samples (0.1 and 0.2), mild samples (0.4), and high performing samples (0.8 and 1.6). The classification results indicated that FTGML had good activity at 0.8 mg/mL and 1.6 mg/mL addition, and was a suitable anti-melanin agent and antioxidant. The 10 indicators were systematically clustered, and the results are shown in Figure 6B-2, which can be divided into two major categories and four minor categories, which is consistent with what is shown in Figure 6A. Among the subcategories, the antioxidant indicators that were in the same category as the anti-melanin indicators were MDA and ROS.

The correlation between anti-melanin properties and other antioxidant properties is shown in Figure 6C. Melanin content was positively correlated with cAMP and tyrosinase activity (both greater than 0.85), indicating a strong correlation between these three factors. GSH and ROS were negatively/positively correlated with melanin (−0.90 for GSH and 0.96 for ROS), indicating that an increase in GSH and a decrease in ROS favored a reduction in melanin content, which was related to the free radical scavenging capacity of GSH, and the fact that a decrease in ROS inhibited the melanin production process. MDA was positively correlated with melanin (0.95), suggesting that a reduction in the content of lipid peroxidation products in the cell membrane favors a reduction in melanin content. A strong correlation between MDA and ROS (0.99), and a negative correlation between CAT and melanin (−0.95), suggest that reactive oxygen species affecting the cell can directly regulate melanin production. Overall, GSH, CAT, MDA, and ROS are the main antioxidant factors influencing melanin production.

## 4. Conclusions

This study investigated the effects of tyrosinase inhibitory peptide FTGML, derived from grass carp fish scale gelatin, on the melanin inhibitory signaling pathway and intracellular antioxidant activity in murine B16F10 melanoma cells. The results showed that FTGML significantly inhibited intracellular tyrosinase activity and melanin content, and had a positive effect on intracellular antioxidant activity. There was a strong correlation between MDA, ROS, GSH, and CAT among the antioxidant indicators for their ability to counteract melanin, which provides theoretical support for predicting the anti-melanin ability or antioxidant activity. In addition, two signaling pathways through which FTGML affects melanin synthesis were revealed. FTGML downregulates MITF expression by inhibiting the cAMP-PI3K/Akt signaling pathway, as well as p38 and JNK in the MAPK signaling pathway. On the other hand, p38 and JNK were found to be present for the first time in the peptide mediated melanin synthesis signaling pathway, whereas this signaling pathway and the STAT3 factors also promoted apoptosis in B16F10 cells. These results suggested that FTGML can reduce melanin production in mouse B16F10 melanoma cells.

## Figures and Tables

**Figure 1 foods-11-00391-f001:**
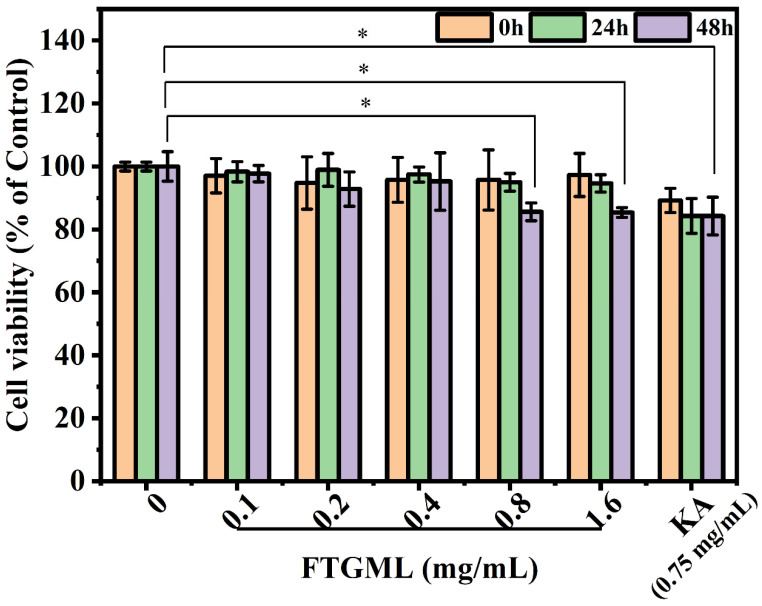
Effects of FTGML and kojic acid (KA) on cell viability. The CCK8 method was used to detect the effects of FTGML at 0, 0.1, 0.2, 0.4, 0.8, and 1.6 mg/mL; and kojic acid at 0.75 mg/mL on the activity of B16F10 cells (Note: *, *p* < 0.05).

**Figure 2 foods-11-00391-f002:**
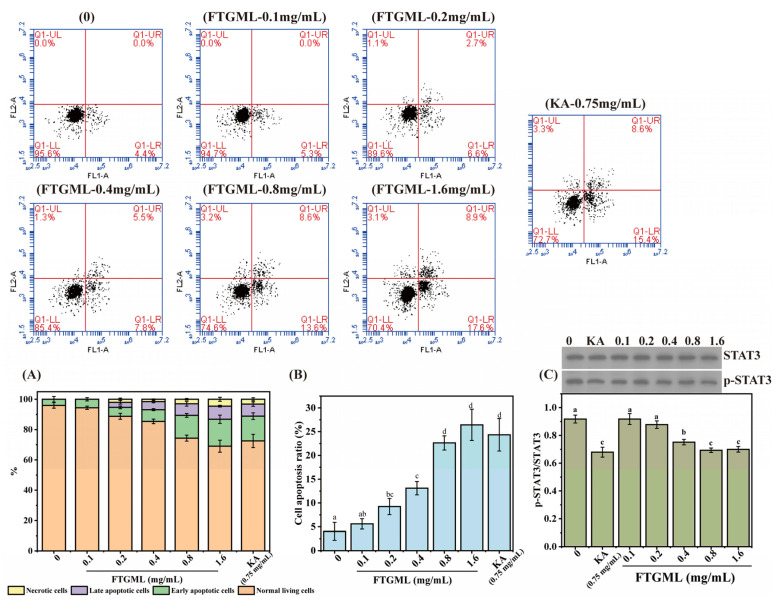
Effects of FTGML and kojic acid (KA) on cell apoptosis. (**A**) The proportion of different states (normal cells, early apoptotic cells, late apoptotic cells, and necrotic cells). (**B**) The apoptosis rates (the sum of the proportions of early and late apoptosis) of B16F10 cells at different treatments. (**C**) The effect of FTGML and kojic acid on the expression of STAT3. Means with different lowercase letters are significantly different (*p* < 0.05) among the different groups.

**Figure 3 foods-11-00391-f003:**
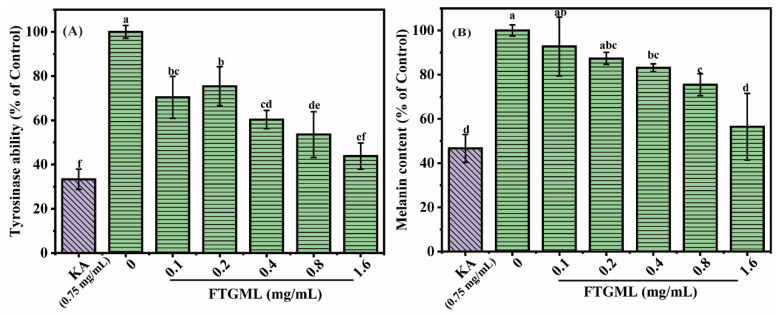
Effects of FTGML and kojic acid (KA) on melanin content and tyrosinase activity in B16F10 melanoma cells. Cells were treated with various concentrations for 48 h (0–1.6 mg/mL). (**A**) Tyrosinase activity. (**B**) Relative melanin content. Means with different lowercase letters are significantly different (*p* < 0.05) among the different groups.

**Figure 4 foods-11-00391-f004:**
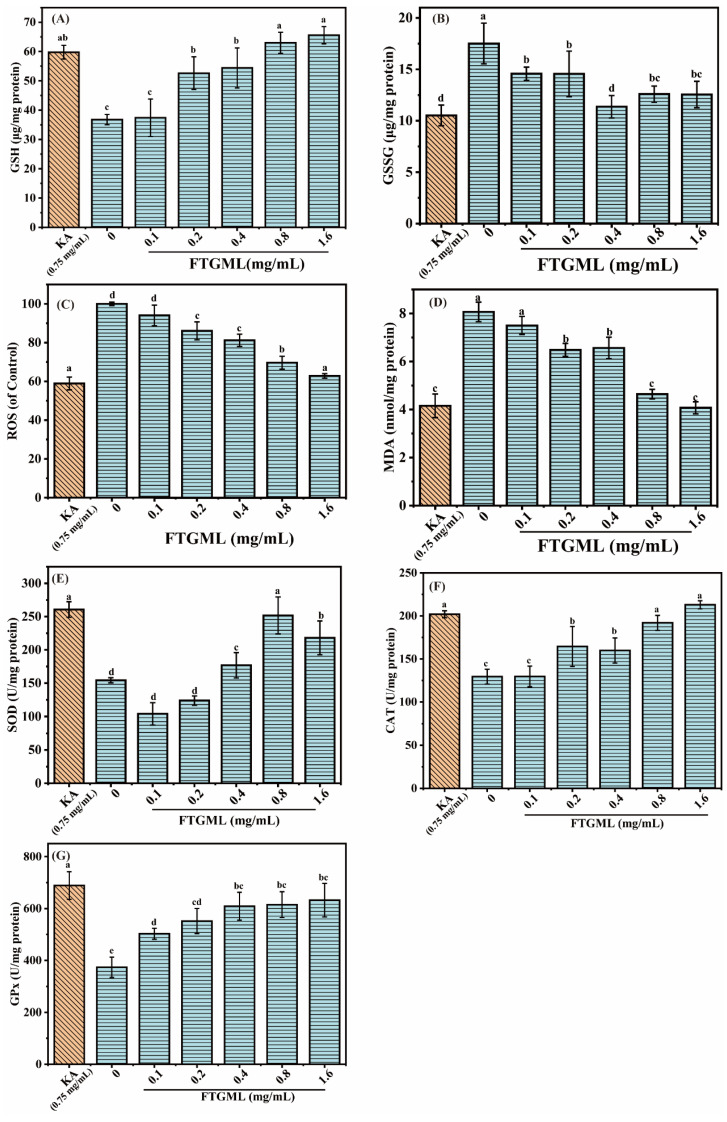
The effect of FTGML and kojic acid on intracellular antioxidant activity. The GSH contents (**A**), GSSG contents (**B**), ROS levels (**C**), MDA contents (**D**), SOD (**E**), CAT (**F**), and GPX (**G**) activities of B16F10 cells at different treatments. Means with different lowercase letters are significantly different (*p* < 0.05) among the different groups.

**Figure 5 foods-11-00391-f005:**
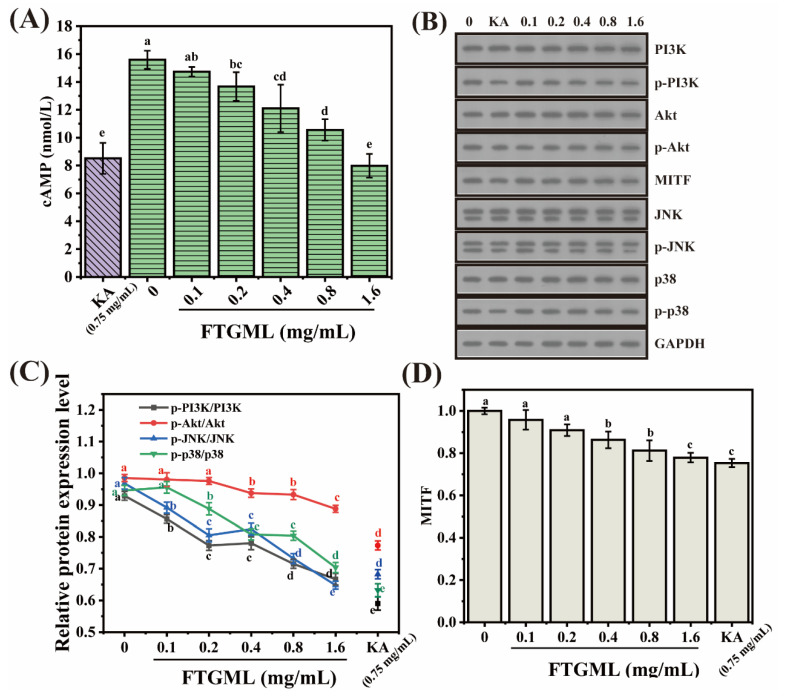
The effect of different concentrations of FTGML on the expression level of related proteins. 0.1, 0.2, 0.4, 0.8, and 1.6 represent the concentration of FTGML (mg/mL), KA represents kojic acid, and the concentration is 0.75 mg/mL. The cAMP contents (**A**), western blot analysis. (**B**), the relative protein expression was examined by western blot analysis (**C**), the relative protein expression of MITF was examined by western blot analysis (**D**). Means with different lowercase letters are significantly different (*p* < 0.05) among the different groups.

**Figure 6 foods-11-00391-f006:**
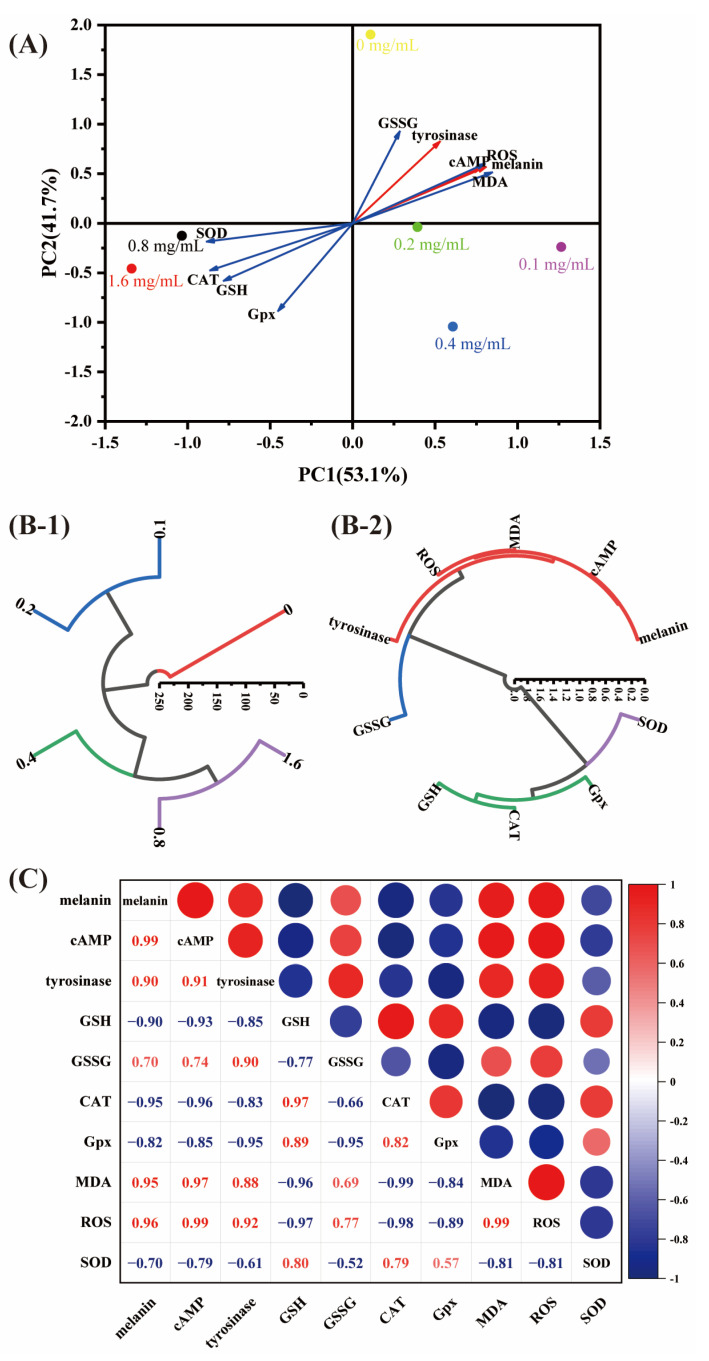
Principal component analysis (**A**) loading plot and score plot. (**B-1**) Hierarchical cluster analysis under different FTGML additions. (**B-2**) Hierarchical cluster analysis of various indicators under different FTGML additions. (**C**) Results of correlation analysis. melanin: mean melanin content (% of control); cAMP: mean cAMP content (nmol/L); tyrosinase: mean tyrosinase ability (% of control); GSH and GSSG: mean GSH and GSSG content (μg/mg protein), respectively; MDA: mean MDA content (nmol/mg protein); ROS: mean ROS content (IU/mL); CAT, GPx, and SOD: mean CAT, GPx, and SOD content (μg/mg protein), respectively.

**Table 1 foods-11-00391-t001:** Characteristic values and contribution rates of principal components.

Items	Variance of Initial Eigenvalues			Extract Square Sum Load Variance		
	Eigenvalue	Variance Contribution Rate/%	Accumulative Contribution Rate/%	Eigenvalue	Variance Contribution Rate/%	Accumulative Contribution Rate/%
Principal component 1	8.714	87.144	87.144	5.311	53.110	53.110
Principal component 2	0.769	7.692	94.836	4.173	41.727	94.836
Principal component 3	0.325	3.246	98.083			
Principal component 4	0.140	1.401	99.484	−		
Principal component 5	0.052	0.516	100.000			
Principal component 6	2.834 × 10^−16^	2.834 × 10^−15^	100.000			
Principal component 7	4.313 × 10^−17^	4.313 × 10^−16^	100.000			
Principal component 8	−3.498 × 10^−17^	−3.498 × 10^−16^	100.000			
Principal component 9	−1.600 × 10^−16^	−1.600 × 10^−15^	100.000			
Principal component 10	−2.691 × 10^−16^	−2.691 × 10^−15^	100.000			

**Table 2 foods-11-00391-t002:** Eigenvectors and load matrices corresponding to principal components.

Items	Principal Component 1		Principal Component 2	
	Feature Vector	Load	Feature Vector	Load
melanin	0.154	0.778	−0.011	0.562
cAMP	0.170	0.807	−0.024	0.567
tyrosinase	−0.154	0.529	0.342	0.824
GSH	−0.146	−0.781	−0.001	−0.580
GSSG	−0.366	0.285	0.567	0.926
CAT	−0.259	−0.863	0.130	−0.475
Gpx	0.238	−0.451	−0.436	−0.884
MDA	0.225	0.845	−0.089	0.513
ROS	0.147	0.800	0.005	0.599
SOD	−0.442	−0.884	0.372	−0.185

**Table 3 foods-11-00391-t003:** Principal component score, comprehensive score, and ranking.

Dosage of FTGML (mg/mL)	Principal Component 1 Score (F1)	Principal Component 2 Score (F2)	Comprehensive Score (F)	Ranking
0	0.10872	1.90461	85.25	6
0.1	1.26466	−0.23872	57.20	5
0.2	0.39347	−0.03859	19.29	4
0.4	0.60744	−1.04331	−11.27	3
0.8	−1.03487	−0.12622	−60.23	2
1.6	−1.33942	−0.45777	−90.24	1

## Data Availability

The data presented in this study are available on request from the corresponding author.

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
