# Peer review of "Effect of Grass Carp Scale Collagen Peptide FTGML on cAMP-PI3K/Akt and MAPK Signaling Pathways in B16F10 Melanoma Cells and Correlation between Anti-Melanin and Antioxidant Properties"

_foods, 2022, doi:10.3390/foods11030391_

Round 1
Reviewer 1 Report
In this study, the authors investigated how a fish-scale-derived peptide FTGML could serve as an anti-melanogenic agent. In particular, evidence of its effect on the apoptosis induction in B16F10 melanoma cells, melanin production, as well as potential contribution of the cellular antioxidant defense and the modulation of signaling pathways has been presented. Strength of the study is that it can be considered a relatively comprehensive investigation, presenting not only cellular evidence, but also biochemical and molecular evidence.
I found no very major/critical issues in the paper. But there are a few places where information presented can be made clearer/more complete/accurate. Below are a number of suggestions I have for the authors’ consideration when revising.
- Title – it may need to be rephrased. It sounds like the peptide could down-regulate correlation (?!)
- Line 37 – Please introduce the abbreviations.
- Line 81 – “and provided guidance for predicting its anti-melanin potential through antioxidant activity in the future.” – Guidance? The meaning of this statement is unclear.
- Lines 98-106 – “3. Cell viability assay”
- Based on Figure 1, the authors presented data for 48 hours. But here, there is no mention of that. Could the authors please check again?
- There is no mention of the addition of the peptide and kojic acid (and how they were dissolved/added) in this section. Could the authors please provide the information?
- Lines 116-125 – “6. Measurement of melanin synthesis”
- Was the detection of melanin just based on absorbance reading at 405 nm? Is that specific enough for melanin detection?
- Lines 126-132 – “7. Assay of Antioxidant activity”
- Could the authors please indicate the duration of treatment before the cells were sampled for antioxidant activity measurements?
- If the cells were sampled from the same cultures as that prepared for the melanin measurement, then it may be clearer to combine 2.6 and 2.7.
- Lines 142-156 – “9. Western blot analysis”
- Way of writing could be improved. The current one sounds like it is giving instructions.
- “B16F10 cells were treated with FTGML (0-1.6 mg/mL) or kojic acid (0.75 mg/mL).” - Please indicate the treatment duration.
- Please indicate briefly how “protein was quantified” and cite reference(s) where relevant.
- Lines 175-178 – “FTGML exhibited lower cytotoxicity than kojic acid when the concentration of FTGML (0.8mg/mL and 1.6mg/mL) was higher than that of kojic acid (0.75mg/mL) … FTGML is safer than kojic acid.”
- Actually, looking at the Figure 1, because of the large error bars, I am not truly convinced that there is any statistically significant difference between 0.8 mg/mL and 1.6 mg/mL versus kojic acid. If statistically significant differences cannot be established, then it is inaccurate to say that “FTGML is safer than kojic acid” or to say “FTGML exhibited lower cytotoxicity than kojic acid”. It could be just similarly/equally safe or cytotoxic as kojic acid.
- Unfortunately, there is no indication of statistical significance in Figure 1 (unlike, e.g., Figures 2B & 2C). So, it would be useful if the authors could include such statistical info in Figure 1 and then interpret the results taking into consideration the statistics.
- Also, could the authors briefly explain why they chose the concentration of 0.75 mg/mL kojic acid, instead of 0.8 mg/mL, which could make comparison between FTGML and kojic acid even more straightforward?
- Line 192 – “The effect of 0.1~1.6 mg/mL FTGML treatment for 24 h on the apoptosis…” – Could the authors briefly indicate why they chose 24 h as treatment duration in apoptosis analysis, but 48 hours in others, e.g., tyrosinase activity and melanin content (Figure 3)? Wouldn’t using the same duration make comparison/connecting different results more straightforward?
- Figure 2:
- Part (c) is somewhat confusing because the orders of treatments differ between the gel images and the bar chart.
- Please recheck whether “STST3” is correct.
- Line 218 – “… silencing STAT3 can significantly inhibit the proliferation of melanoma cells and promote cell apoptosis” – Based on Figure 2C (STAT3 activation), the difference between 0.4 and 0.8 mg/mL FTGML seems minor, but in Figure 2B (apoptosis), the difference between the two treatments seems quite drastic. Based on this, could it be that the role of STAT3 silencing is not that crucial in the induction of apoptosis by FTGML treatment?
- Lines 238-239 – “The results in Figure 3B show that FTGML significantly reduced the melanin of B16F10 cells. The intracellular melanin content gradually decreased with the increase of FTGML concentration.” – These statements seem inaccurate and should be revised/elaborated. In Figure 3B, there is no statistical difference between the melanin levels in 0, 0.1, 0.2, 0.4 mg/mL FTGML treated cells. In other words, up to 0.4 mg/mL FTGML, there was no statistically significant reduction in melanin contents in the cells.
- Could the authors provide a possible explanation for this observation - when there was very clear (statistically significant) reduction in tyrosinase activity starting from 0.1 to 0.4 mg/mL FTGML (Fig 3A), there was no clear (statistically significant) change in melanin content over the same concentration range?
- For the GSH and GSSG results (Figure 4), did the authors also consider changes in GSG:GSSG ratios? In Figure 4B, there is a sudden dip in GSSG level in the 0.4 mg/mL FTGML treatment. Would the trend become clearer if the result is presented as GSH:GSSG ratios?
- Lines 299-300 – “FTGML shows a protective effect on oxidation by increasing the activity of antioxidant enzymes in B16F10 cells, thereby avoiding melanin deposition.” – Could the authors please clarify whether both results were obtained from the 48h-treated cells? As mentioned above, for antioxidant assays, the duration of treatment was not indicated clearly in M&M. If both results were not obtained from cells treated for the same duration, then the connection between them as indicated in the statement seems less strong.
- Line 320 – “… (p > 0.05) … ” – Is this information correct? Please recheck.
- The authors investigated different signaling pathways – could the results obtained pinpoint which one is the more crucial one whose modulation following FTGML treatment led to apoptosis and melanogenesis?
- Line 463 – “Considering the low cytotoxicity of FTGML and its similar effect to that of kojic acid …” – This statement may need to be rephrased to make it more specific/accurate what the “similar effects” refers to. Also, please recheck whether after considering statistical significance in Figure 1, is the cytotoxicity of FTGML only similar/comparable to, but not lower than, kojic acid?
Author Response
In this study, the authors investigated how a fish-scale-derived peptide FTGML could serve as an anti-melanogenic agent. In particular, evidence of its effect on the apoptosis induction in B16F10 melanoma cells, melanin production, as well as potential contribution of the cellular antioxidant defense and the modulation of signaling pathways has been presented. Strength of the study is that it can be considered a relatively comprehensive investigation, presenting not only cellular evidence, but also biochemical and molecular evidence.
Response: Thanks for your affirmation of our work.
I found no very major/critical issues in the paper. But there are a few places where information presented can be made clearer/more complete/accurate. Below are a number of suggestions I have for the authors’ consideration when revising.
- Title – it may need to be rephrased. It sounds like the peptide could down-regulate correlation (?!)
Response: Thanks for your suggestion. We have reframed the title.
- Line 37 – Please introduce the abbreviations.
Response: Thanks for your suggestion. We have added details to these abbreviations in revised manuscript.
- Line 81 – “and provided guidance for predicting its anti-melanin potential through antioxidant activity in the future.” – Guidance? The meaning of this statement is unclear.
Response: Thanks for your suggestion, we have amended this sentence so that the meaning of the sentence is clearer.
- Lines 98-106 – “3. Cell viability assay”
- Based on Figure 1, the authors presented data for 48 hours. But here, there is no mention of that. Could the authors please check again?
- There is no mention of the addition of the peptide and kojic acid (and how they were dissolved/added) in this section. Could the authors please provide the information?
Response: Thanks for your suggestion. You are right, the “48 hours” was missing in previous sentence and we have added it. Furthermore, we have added details according to your suggetion in this section.
- Lines 116-125 – “6. Measurement of melanin synthesis”
- Was the detection of melanin just based on absorbance reading at 405 nm? Is that specific enough for melanin detection?
Response: Melanin has a strong absorption peak at 405nm. Pulished papers have proved that when melanin secretion by B16F10 cells is inhibited by the sample, the absorbance value decreases and the melanogenesis inhibition rate can be calculated by this method. We also cited one published paper in section 2.6. Related papers are as follows:
Zhang X, Li J, Li Y, et al. Anti-melanogenic effects of epigallocatechin-3-gallate (EGCG), epicatechin-3-gallate (ECG) and gallocatechin-3-gallate (GCG) via down-regulation of cAMP/CREB /MITF signaling pathway in B16F10 melanoma cells [J]. Fitoterapia, 2020, 145: 104634.
Huang H C, Chou Y C, Wu C Y, et al. [8]-Gingerol inhibits melanogenesis in murine melanoma cells through down-regulation of the MAPK and PKA signal pathways [J]. Biochem. Bioph. Res. Co., 2013, 438(2): 375-381.
- Lines 126-132 – “7. Assay of Antioxidant activity”
- Could the authors please indicate the duration of treatment before the cells were sampled for antioxidant activity measurements?
- If the cells were sampled from the same cultures as that prepared for the melanin measurement, then it may be clearer to combine 2.6 and 2.7.
Response: Thanks for your suggestion. You are right, we did use the samples from section 2.6 to determine the antioxidant activity. We have added the relevant explanation in section 2.7.
- Lines 142-156 – “9. Western blot analysis”
- Way of writing could be improved. The current one sounds like it is giving instructions.
- “B16F10 cells were treated with FTGML (0-1.6 mg/mL) or kojic acid (0.75 mg/mL).” - Please indicate the treatment duration.
- Please indicate briefly how “protein was quantified” and cite reference(s) where relevant.
Response: Thanks for your suggestion. We have rewritten this section to make it more readable and also corrected some mistakes according to your advice.
- Lines 175-178 – “FTGML exhibited lower cytotoxicity than kojic acid when the concentration of FTGML (0.8mg/mL and 1.6mg/mL) was higher than that of kojic acid (0.75mg/mL) … FTGML is safer than kojic acid.”
- Actually, looking at the Figure 1, because of the large error bars, I am not truly convinced that there is any statistically significant difference between 0.8 mg/mL and 1.6 mg/mL versus kojic acid. If statistically significant differences cannot be established, then it is inaccurate to say that “FTGML is safer than kojic acid” or to say “FTGML exhibited lower cytotoxicity than kojic acid”. It could be just similarly/equally safe or cytotoxic as kojic acid.
- Unfortunately, there is no indication of statistical significance in Figure 1 (unlike, e.g., Figures 2B & 2C). So, it would be useful if the authors could include such statistical info in Figure 1 and then interpret the results taking into consideration the statistics.
- Also, could the authors briefly explain why they chose the concentration of 0.75 mg/mL kojic acid, instead of 0.8 mg/mL, which could make comparison between FTGML and kojic acid even more straightforward?
Response: Thanks for your suggestion. You are right, we made a wrong statement, and 0.8mg/mL and 1.6mg/mL didn’t show any statistically significant difference. In fact, the main purpose of this section of the experiment is to demonstrate that FTGML is safe to B16F10 cells within the specified concentration range, which will serve as the foundation for later experiments. And we have deleted this statement in revised manuscript.
The statistical analysis of the data in Figure 1 showed some significant differences (p < 0.05) after 48 hours duration and some details were added in Fig. 1. These significant differences could explain the slight cytotoxicity effect (cell viability > 80%) of increasing concentration.
Regarding 0.75 mg/mL kojic acid was chosen as the positive control. In this study, the use of kojic acid as a positive control is to qualitatively compare the difference between FTGML and the existing mature whitening ingredients. However, a more comprehensive comparison should be avoided, because this article is not specifically comparing the difference between different concentrations of FTGML and kojic acid. As a result, we did a simple preliminary experiment in the early stage, using the kojic acid concentration of 0, 0.25, 0.5, 0.75 and 1 mg/mL, which covered the recommended values of kojic acid in most published similar papers, and we checked the cell viability at these concentrations. It's better to choose a higher kojic acid concentration as much as possible to make the effect by kojic acid more meaningful, but we found cell viability < 80% at 1 mg/mL, therefore, we chose 0.75 mg/L as a positive control.
- Line 192 – “The effect of 0.1~1.6 mg/mL FTGML treatment for 24 h on the apoptosis…” – Could the authors briefly indicate why they chose 24 h as treatment duration in apoptosis analysis, but 48 hours in others, e.g., tyrosinase activity and melanin content (Figure 3)? Wouldn’t using the same duration make comparison/connecting different results more straightforward?
Response: Thanks for your suggestion. This was a mistake, and we did choose 48 h for apoptosis analysis. We have corrected it.
- Figure 2:
- Part (c) is somewhat confusing because the orders of treatments differ between the gel images and the bar chart.
- Please recheck whether “STST3” is correct.
Response: Thanks for your suggestion. We have corrected these two mistakes in revised manuscript.
- Line 218 – “… silencing STAT3 can significantly inhibit the proliferation of melanoma cells and promote cell apoptosis” – Based on Figure 2C (STAT3 activation), the difference between 0.4 and 0.8 mg/mL FTGML seems minor, but in Figure 2B (apoptosis), the difference between the two treatments seems quite drastic. Based on this, could it be that the role of STAT3 silencing is not that crucial in the induction of apoptosis by FTGML treatment?
Response: Thanks for your suggestion. Duncan test was used for analyzing the significantly difference and the result showed that it did exist difference between 0.4 and 0.8 mg/L in Figure 2C despite it seemed minor. On the other hand, the difference between 0.4 and 0.8 mg/L was more significant in Figure 2B than 2C, which was thought to be due to the different detection methods of the two images. Data in Figure 2B was detected by a flow cytometer, while that in 2C was used the Western Blot method and an image processing software. Furthermore, the indicators of Figure 2B and 2C are also not linear. As above influencing factors, the trend of significant difference between 0.4 and 0.8 mg/L in the two figures could be different. According to these results, it can be seen that STAT3 silencing is still important in the induction of apoptosis by FTGML treatment.
- Lines 238-239 – “The results in Figure 3B show that FTGML significantly reduced the melanin of B16F10 cells. The intracellular melanin content gradually decreased with the increase of FTGML concentration.” – These statements seem inaccurate and should be revised/elaborated. In Figure 3B, there is no statistical difference between the melanin levels in 0, 0.1, 0.2, 0.4 mg/mL FTGML treated cells. In other words, up to 0.4 mg/mL FTGML, there was no statistically significant reduction in melanin contents in the cells.
Response: Thanks for your suggestion. You are right and this was a wrong statement. We have rewritten this part in revised manuscript.
- Could the authors provide a possible explanation for this observation - when there was very clear (statistically significant) reduction in tyrosinase activity starting from 0.1 to 0.4 mg/mL FTGML (Fig 3A), there was no clear (statistically significant) change in melanin content over the same concentration range?
Response: Thanks for your suggestion. The activity of tyrosinase is just one of many factors that affect the amount of melanin. Other factors, such as antioxidant properties, could lead to changes in melanin content if co-work with tyrosinase. However, as demonstrated in Figure 4, the change in antioxidant capacity is minimal at low FTGML concentrations. The low FTGML concentrations could result in a decrease only in tyrosinase activity, which can not lead a considerable reduction in melanin content.
On the other hand, we found that the error bar in these two figures were wrong, because the tyrosinase activity and the melanin content were both converted and displayed as “% of control”, but the error bar was still calculated using the data before. So we corrected this error.
- For the GSH and GSSG results (Figure 4), did the authors also consider changes in GSG:GSSG ratios? In Figure 4B, there is a sudden dip in GSSG level in the 0.4 mg/mL FTGML treatment. Would the trend become clearer if the result is presented as GSH:GSSG ratios?
Response: Thanks for your suggestion. We have added the GSG:GSSG ratios in text part according to your advice.
- Lines 299-300 – “FTGML shows a protective effect on oxidation by increasing the activity of antioxidant enzymes in B16F10 cells, thereby avoiding melanin deposition.” – Could the authors please clarify whether both results were obtained from the 48h-treated cells? As mentioned above, for antioxidant assays, the duration of treatment was not indicated clearly in M&M. If both results were not obtained from cells treated for the same duration, then the connection between them as indicated in the statement seems less strong.
Response: Thanks for your suggestion. We have added the related description in revised manuscript.
- Line 320 – “… (p > 0.05) … ” – Is this information correct? Please recheck.
Response: Thanks for your suggestion. We have corrected this mistake.
- The authors investigated different signaling pathways – could the results obtained pinpoint which one is the more crucial one whose modulation following FTGML treatment led to apoptosis and melanogenesis?
Response: Thanks for your suggestion. Melanin is affected by many signal pathways. In this article, the PI3K signal pathway is the most crucial signal pathway after FTGML treatment. We have added this conclusion in the revised manuscript.
- Line 463 – “Considering the low cytotoxicity of FTGML and its similar effect to that of kojic acid …” – This statement may need to be rephrased to make it more specific/accurate what the “similar effects” refers to. Also, please recheck whether after considering statistical significance in Figure 1, is the cytotoxicity of FTGML only similar/comparable to, but not lower than, kojic acid?
Response: Thanks for your suggestion. We rephrased this sentence. The cytotoxicity of FTGML does not need to be compared with kojic acid. The purpose is to demonstrate that the concentration range chosen is available, and addition dose causing cellular activity >80% is generally considered to be safe and usable.
Reviewer 2 Report
In this manuscript the authors evaluated the effects of a tyrosine inhibitor peptide FTGML, obtained from grass carp fish gelatin in a murine B6F10 melanoma cell line.
In my opinion this manuscript needs a major revision. So the authors must to improve the manuscript quality and correct it according to the following comments:
Why was only used one cancer cell line? And why this cell line? Why not a humane melanoma cell line?
What are the effects of FTGML in normal skin cell lines? This results should be added to the manuscript.
What is the composition of FTGML? Particularly the composition of the product used in this research. Why were used these concentrations?
The last phrase of conclusions:"Considering the low cytotoxicity of FTGML and its similar effect to that of kojic acid, we suggest that FTGML could be used as a natural decolourising agent in health food, cosmetics and pharmaceuticals."
Are the authors really sure about this sentence? Do they think that with the results obtained here one can really move on to this type of application?
Author Response
In this manuscript the authors evaluated the effects of a tyrosine inhibitor peptide FTGML, obtained from grass carp fish gelatin in a murine B6F10 melanoma cell line.
In my opinion this manuscript needs a major revision. So the authors must to improve the manuscript quality and correct it according to the following comments:
Why was only used one cancer cell line? And why this cell line? Why not a humane melanoma cell line?
Response: Murine B16F10 cells have the advantages of multiple passages, rapid development and relatively simple requirements for culture conditions. They are the first choice for screening whitening active ingredients in many published papers. Therefore, the tests in this paper were performed on murine B16F10 cells. Another choice of tumor-derived cell is the human epidermal melanoma A375 cells. The above two cells are different in physiological and biochemical reactions from normal human melanocytes, and they are only used for large-scale or high-throughput preliminary screening of whitening agents. Thanks very much for your suggestions. We will also explore the effect of FTGML on human epidermal melanoma cells in future studies.
What are the effects of FTGML in normal skin cell lines? This results should be added to the manuscript.
Response: Thanks for your suggestion. The purpose of this study is to explore the influence of FTGML on the melanogenesis signal pathway, so murine B16F10 was chosen because it was reliable and convenient for the test. What you said about the impact in normal cell lines is also the direction we are currently trying to explore. Thanks again for your suggestions.
What is the composition of FTGML? Particularly the composition of the product used in this research. Why were used these concentrations?
Response: FTGML is a pentapeptide whose amino acid composition is Phe-Thr-Gly-Met-Leu. This peptide was screened out in our previous research and found to have a good whitening effect, so we explored its possible signal pathways in this research. The selected concentration range is obtained by referring to relevant literature and preliminary experiments.
The last phrase of conclusions:"Considering the low cytotoxicity of FTGML and its similar effect to that of kojic acid, we suggest that FTGML could be used as a natural decolourising agent in health food, cosmetics and pharmaceuticals."
Are the authors really sure about this sentence? Do they think that with the results obtained here one can really move on to this type of application?
Response: Thanks for your suggestion. You are right. Strictly speaking, our statement in conclusion was incorrect, because there are still many unknown influencing factors before this application is reached, and we have made revision in this part.
Round 2
Reviewer 2 Report
The authors did not improve the manuscript scientific quality. They merely replied to my comments evasively and did not make manuscript corrections/revision. Most of the replies were made because they felt like it.
What is the point of assessing the effect of a substance if even the composition of the substance has not been done? What is the interest in answering: we are thinking about it for future work? This manuscript has serious scientific flaws that have not been improved.
Author Response
The authors did not improve the manuscript scientific quality. They merely replied to my comments evasively and did not make manuscript corrections/revision. Most of the replies were made because they felt like it.
What is the point of assessing the effect of a substance if even the composition of the substance has not been done? What is the interest in answering: we are thinking about it for future work? This manuscript has serious scientific flaws that have not been improved.
Response: We are sorry that the previous modification and reply did not meet your expectations. We have furtherly modified our manuscript according to your comments. Moreover, all comments including this time and last time have been answered in detail. Thanks for your attention in our manuscript.
Regarding the composition of FTGML, we are sorry about the less information described FTGML before. In the revised manuscript, we have added a description of FTGML in the “Introduction”. FTGML is a new peptide derived from grass carp fish scale gelatin hydrolysate in our previous work. In previous work, we screened the peptides with high tyrosinase inhibitory activity in grass carp fish scale gelatin hydrolysate, and furtherly used mass spectrometry to identify them, which showed that FTGML had the highest tyrosinase inhibitory activity. FTGML is composed of 5 amino acids with phenylalanine-threonine-glycine-methionine-leucine (FTGML). And we also conducted animal experiments on FTGML in the previous work to verify its anti-melanin ability. These preliminary work provided the foundation for this manuscript.
Regarding the thinking about future work, we believe that your statement is appropriate considering your previous comment. Due to the time limitation of the last revision (only 10 days), we were unable to perform this experiment. After receiving your comments this time, we decided to conduct this experiment right away. When constructing the experimental program, we found some problems of selection of human melanoma cells and human normal skin cells, which we need to explain to you.
1) More strict conditions are required in human cell culture and processing, as well as to further cultivate human melanoma cells, thus many research employ mouse cells. The mouse melanoma cell line B16F10 is extremely consistent with human melanoma cells in terms of cell structure and melanin synthesis, and with the advantages of many passages, rapid development, relatively simple culture conditions, high malignancy, and good tumorigenicity. In addition, the experimental results will be incomparable if using different cell lines, due to differences in protein or receptor expression levels between them. The selection of a commonly used cell line for study provides a clearer and representative understanding of its mechanism of action. As a result, we preserved the mouse melanoma cells B16F10 and did not supplement the experiments with human melanoma cells.
2) The melanin content in normal human skin cells is quite low, making it difficult to observe the change of melanin content after inhibitor treatment. Melanin must be enriched to get the requisite amount in human normal skin cells, while normal cells may become cancer cell in the enrichment. Therefore, human normal skin cells are usually not selected for anti-melanin related experiments in many studies. We provide a more detailed explanation below and hope for your understanding.
Reviewer’s comments in last time:
In this manuscript the authors evaluated the effects of a tyrosine inhibitor peptide FTGML, obtained from grass carp fish gelatin in a murine B6F10 melanoma cell line.
In my opinion this manuscript needs a major revision. So the authors must to improve the manuscript quality and correct it according to the following comments:
Why was only used one cancer cell line? And why this cell line? Why not a humane melanoma cell line?
Response: The experimental results will be incomparable if using different cell lines, due to differences in protein or receptor expression levels between them. The selection of a commonly used cell line for study provides a clearer and representative understanding of its mechanism.
In terms of cell structure and melanin synthesis, the mouse melanoma cell line B16F10 is highly consistent with human melanoma cell. Furthermore, it is difficult to culture human melanoma cell when evaluating the efficacy of whitening active due to its more strict requirements for cell culture and handling condition. The mouse melanoma cell line B16F10 has the advantages of multiple passages, rapid development, relatively simple culture conditions, high malignancy and good tumorigenicity. So B16F10 cells are widely used as the effective cells for cell evaluation of whitening active substances [1-7]. In many published papers, they are the first choice for screening whitening active ingredients.
- Ohno O, Watabe T, Nakamura K, et al. Inhibitory effects of bakuchiol, bavachin, and isobavachalcone isolated from Piper longum on melanin production in B16 mouse melanoma cells [J]. Biosci Biotechnol Biochem, 2010, 74(7): 1504-1506.
- Kubo I, Nihei K, Tsujimoto K. Methyl p-coumarate, a melanin formation inhibitor in B16 mouse melanoma cells [J]. Bioorg Med Chem, 2004, 12(20): 5349-5354.
- Usuki A, Ohashi A, Sato H, et al. The inhibitory effect of glycolic acid and lactic acid on melanin synthesis in melanoma cells [J]. Exp Dermatol, 2003, 12: 43-50.
- Zhang X, Li J, Li Y, et al. Anti-melanogenic effects of epigallocatechin-3-gallate (EGCG), epicatechin-3-gallate (ECG) and gallocatechin-3-gallate (GCG) via down-regulation of cAMP/CREB /MITF signaling pathway in B16F10 melanoma cells [J]. Fitoterapia, 2020, 145: 104634.
- Han J H, Bang J S, Choi Y J, et al. Anti-melanogenic effects of oyster hydrolysate in UVB-irradiated C57BL/6J mice and B16F10 melanoma cells via downregulation of cAMP signaling pathway [J]. J. Ethnopharmacol., 2019, 229: 137-144.
- Azam M S, Kwon M, Choi J, et al. Sargaquinoic acid ameliorates hyperpigmentation through cAMP and ERK-mediated downregulation of MITF in α-MSH-stimulated B16F10 cells [J]. Biomed. Pharmacother., 2018, 104: 582-589.
- Han J S, Sung J H, Lee S K. Antimelanogenesis Activity of Hydrolyzed Ginseng Extract (GINST) via Inhibition of JNK Mitogen-activated Protein Kinase in B16F10 Cells [J]. J. Food Sci., 2016, 81(8): H2085-H2092.
What are the effects of FTGML in normal skin cell lines? This results should be added to the manuscript.
Response: Thank you for your suggestions. We believe that your statement is appropriate considering your previous comment. Due to the time limitation of the last revision (only 10 days), we were unable to perform this experiment. After receiving your comments this time, we decided to conduct this experiment right away. When formulating the plan, we reviewed a lot of relevant literature and consulted many researchers in this field, but the feedback we got was that it was not suitable for selecting normal skin cells for this research. The main reasons are as follows: The melanin content in normal skin cells is very low, a certain amount of melanin accumulation needs to be achieved to satisfy the observation of melanin changes after the inhibitor treatment. If normal skin cell is required to meet the melanin content that is suitable for experiment, it would take a long time and ask various stimulations of external factors. Under the various stimulation of external factors, it is possible to transform from normal cells to cancer cells [1-4].
- Wang L, Oh J Y, Kim Y S, et al. Anti-Photoaging and Anti-Melanogenesis Effects of Fucoidan Isolated from Hizikia fusiforme and Its Underlying Mechanisms [J]. Mar Drugs, 2020, 18(8): 427.
- Han J H, Bang J S, Choi Y J, et al. Anti-melanogenic effects of oyster hydrolysate in UVB-irradiated C57BL/6J mice and B16F10 melanoma cells via downregulation of cAMP signaling pathway [J]. J. Ethnopharmacol., 2019, 229: 137-144.
- Yuan, X. H., Yao, C., Oh, J. H., Park, C. H., Tian, Y. D., Han, M., ... & Lee, D. H.. Vasoactive intestinal peptide stimulates melanogenesis in B16F10 mouse melanoma cells via CREB/MITF/tyrosinase signaling. Biochem Bioph Res Co, 2016, 477(3), 336-342.
- Oh G W, Ko S C, Heo S Y, et al. A novel peptide purified from the fermented microalga Pavlova lutheri attenuates oxidative stress and melanogenesis in B16F10 melanoma cells [J]. Process Biochem, 2015, 50(8): 1318-1326.
What is the composition of FTGML? Particularly the composition of the product used in this research. Why were used these concentrations?
Response: Thanks very much for your comments about the composition of the substance FTGML. We are sorry that we did not give a detailed description of FTGML. In the new revised manuscript, we have added a description of FTGML in the Introduction.
The concentration range was chosen based on FTGML cell viability tests. The FTGML concentration range we set before were 0, 0.1, 0.2, 0.4, 0.8, 1.6, and 3.2 mg/mL, respectively. After treating B16F10 cells with these concentrations, we found that the cell viability after treatment with 3.2 mg/mL FTGML was only 53%, which could not meet the requirements of subsequent experiments (cell viability > 80%), so we chose 0, 0.1, 0.2 , 0.4, 0.8, 1.6 mg/mL as the experimental ranges.
The last phrase of conclusions: "Considering the low cytotoxicity of FTGML and its similar effect to that of kojic acid, we suggest that FTGML could be used as a natural decolourising agent in health food, cosmetics and pharmaceuticals."
Are the authors really sure about this sentence? Do they think that with the results obtained here one can really move on to this type of application?
Response: Thanks for your suggestions. The sentence in previous manuscript lacked rigorous. In the new manuscript, we have replaced this sentence with "These results suggest that FTGML can reduce melanin production.".
Round 3
Reviewer 2 Report
Regrettably the authors fail to improve the manuscript scientifically.
Authors wrote in material and methods:
"FTGML was obtained from the grass carp fish scale gelatin hydrolysate. In brief, grass carp fish scale gelatin was hydrolyzed by alcalase and gastrointestinal simulate digestion to obtain a mixture of peptides. The peptides with tyrosinase inhibitor activity were screened and identified by bioaffinity ultrafiltration-mass spectrometry, and peptide FTGML showed the best tyrosinase inhibitory activity."
Where are the images of spectrometry results?
Unfortunately, I can not accepted this paper for publication. authors should analyze the following paper:
Oncotarget. 2017 Feb 7; 8(6): 10498–10509. Published online 2017 Jan 2. doi: 10.18632/oncotarget.14443
The B16 series of cell lines are derived from a spontaneous melanoma in C57BL/6 mice [25]. Thus the cells can be implanted back in this murine strain within the context of an intact immune system as a syngeneic model of melanoma [26]. However, it should be recognized that animals provide only an approximation of the reality in humans. Due to differences in skin architecture, mouse melanomagenesis does not exactly phenocopy disease progression in humans (reviewed in [27]). Mice, as heavily hair-covered species, have no need for skin pigmentation, and therefore almost completely lack epidermal melanocytes. Instead, melanocytes are located in dermal hair papilla. Thus, unlike most human melanomas that are of epidermal origin and usually lack pigment, most murine lesions are dermal with very high levels of pigment. In addition, the most commonly used murine melanoma model (B16) does not harbor a BRAF mutation [28]. Accordingly, murine melanomas likely do not recapitulate some of the key features of human melanoma and come with a different set of caveats.
This manuscript needs to be revised by an English native. For instance see:
"B16F10 cells was cultured by the medium of RPMI-1640," Instead of was must be were.